

# Inferring microevolution from museum collections and resampling: lessons learned from *Cepaea*

Małgorzata Ożgo[1], Thor-Seng Liew[2,3,4], Nicole B. Webster[3,4] and Menno Schilthuizen[2,3,4]

[1] Department of Evolutionary Biology, Kazimierz Wielki University, Bydgoszcz, Poland
[2] Institute for Tropical Biology and Conservation, Universiti Malaysia Sabah, Kota Kinabalu, Sabah, Malaysia
[3] Institute Biology Leiden, Leiden University, Leiden, The Netherlands
[4] Endless Forms Group, Naturalis Biodiversity Center, Leiden, The Netherlands

## ABSTRACT

Natural history collections are an important and largely untapped source of long-term data on evolutionary changes in wild populations. Here, we utilize three large geo-referenced sets of samples of the common European land-snail *Cepaea nemoralis* stored in the collection of Naturalis Biodiversity Center in Leiden, the Netherlands. Resampling of these populations allowed us to gain insight into changes occurring over 95, 69, and 50 years. *Cepaea nemoralis* is polymorphic for the colour and banding of the shell; the mode of inheritance of these patterns is known, and the polymorphism is under both thermal and predatory selection. At two sites the general direction of changes was towards lighter shells (yellow and less heavily banded), which is consistent with predictions based on on-going climatic change. At one site no directional changes were detected. At all sites there were significant shifts in morph frequencies between years, and our study contributes to the recognition that short-term changes in the states of populations often exceed long-term trends. Our interpretation was limited by the few time points available in the studied collections. We therefore stress the need for natural history collections to routinely collect large samples of common species, to allow much more reliable hind-casting of evolutionary responses to environmental change.

## INTRODUCTION

Evolutionary processes occurring over time-scales as short as decades or years are increasingly documented, especially in the context of human alterations of the environment, including pollution, harvest, or species introductions (*Hendry, Gotanda & Svensson, 2017*; *Schilthuizen, 2018*). Such rapid evolutionary changes, taking place generation-by-generation, can have important consequences for populations, ecosystems, and human societies (*Palkovacs et al., 2012*; *Kinnison, Hairston & Hendry, 2015*; *Hendry, Gotanda & Svensson, 2017*). The interest in the reciprocal influences between ecology and evolution has been growing recently, and while the general picture of eco-evolutionary dynamics is coming into focus, many questions remain open to investigation (*Hendry, 2013*; *Kinnison,*

Corresponding author
Menno Schilthuizen,
menno.schilthuizen@naturalis.nl

*Hairston & Hendry, 2015*; *Hendry, 2017*). One of the factors limiting progress in this field is the scarcity of data, especially long-term records from natural populations (*Kingsolver & Pfennig, 2007*; *Siepielski, DiBattista & Carlson, 2009*; *Siepielski et al., 2013*). Surprisingly, such time-series appear to be more readily available for fossils than for contemporary populations (*Bell, 2010*). Unlike in other studies such as, e.g., spatial analyses, this lack cannot easily be remedied by applying more effort, as available data are limited to the studies previously initiated and maintained. Notable examples include studies on the Darwin's finches in the Galápagos (*Grant & Grant, 2014*), the peppered moth *Biston betularia* in England (*Cook & Saccheri, 2013*), the damselfly *Ischnura elegans* (*Le Rouzic et al., 2015*) or the polymorphic marine snail *Littorina saxatilis* in Sweden (*Johannesson & Butlin, 2017*). In polymorphic land snails, studies based on annual recording of morph frequencies over extended periods of time were carried out on *Cepaea nemoralis* in England (*Cain, Cook & Currey, 1990*; *Bell, 2010*), and on *Theba pisana* in Australia (*Johnson, 2011*). Long-term monitoring of natural populations requires many years of work in often challenging field conditions, and studies of this kind remain relatively few.

Other sources of long-term data include fisheries reports or trophy measurements (e.g., *Sharpe & Hendry, 2009*; *Shackell et al., 2010*; *Douhard et al., 2016*), and, especially, natural history collections. Natural history collections are exceptional in that they contain "raw data"—the actual specimens, and not only records associated with specimens (*Schilthuizen et al., 2015*), and they often date back 100 years or even more. Their value is increasingly recognized (*Holmes et al., 2016*; *Linck et al., 2016*), but they remain a largely untapped resource. Among museum collections, mollusks have a prominent place. Their beauty has always appealed to naturalists and they have been collected extensively. Because mollusk shells are relatively easy to maintain they constitute an important part of museum collections worldwide (*Vinarski, 2016*; *Breure & Araujo, 2017*).

In this paper we identified several large geo-referenced samples of the land snail *Cepaea nemoralis* in the collection of Naturalis Biodiversity Center in Leiden, the Netherlands, and we resampled these populations in the field in 2010. *Cepaea nemoralis* is a common European land snail species with a broad distribution. It has a distinctive shell polymorphism: its shells can be yellow, pink or brown and bear up to five spiral bands; the mode of inheritance of these characteristics is well established (*Murray, 1975*). The environmental drivers of selection include climate and visually hunting predators, and morph frequency shifts in response to these pressures have been documented (reviews in, e.g., *Cook, 1998*; *Cook, in press*; *Ożgo, 2008*). The system has been utilized extensively, but many questions remain unanswered; the most important among them concern the temporal and spatial patterns of selection. The aim of the present study is to add to the understanding of evolutionary processes occurring in populations of *Cepaea nemoralis* over extended periods of time. The use of museum collections allowed us to gain insight into changes occurring over 95, 69, and 50 years.

## MATERIALS AND METHODS

### Historical baseline data

We used the dry Mollusca collection of Naturalis Biodiversity Center to locate large samples of *Cepaea nemoralis* from the Netherlands with locality information precise enough to make re-sampling possible. We identified three such sets of samples (at the time of writing, these samples had not yet received collection numbers; however, they were held in the special "large *Cepaea* sample" section):

(1) Lobith. Shells collected by H Wolda and students on 20th April 1960 ($N = 368$), 27–28th April 1961 ($N = 398$), and 10–14th April 1962 ($N = 1,657$) at 13 positions along a 400-m section of dike on the river bank along the Rhine (51.859°N 6.085°E). The samples were used for a study on stability of a steep c. 20-m-wide cline located along the section, where unbanded yellow shells increase in frequency at the expense of banded yellow and unbanded "red" shells (*Wolda, 1969*). *Wolda (1969)* reports on a total of 26,230 individuals, and writes that most samples consisted of mostly living snails and a small proportion of dead shells. Since Wolda states that all living snails were returned to their collection localities after scoring, we conclude that the Naturalis samples ($N = 2,423$) consist only of snails that were collected as empty shells. *Wolda (1969)* also reports that the vegetation along the dike grades from a dense river-dune vegetation on sand, rich in nutrition and calcium, to more open river-dune vegetation on coarse, rich, and dry sand with calcium.

(2) Empe. Shells collected by JC Van Heurn in May 1915 ($N = 468$) and July 1951 ($N = 340$) at Empe estate (52.145°N 6.142°E). The sample of 1915 is labelled as follows: "Collected at random in one site on the main road from Zutphen to Voorst, in the Zutphen municipality, directly opposite the manor "Empe". The collection consists of 451 adults and 17 juveniles. […] The habitats are the elm stems as well as the roadside-verges. In the verges there were no nettles or generally tall herbs, but there was a row of elm-bushes, mixed with some hawthorn, grasses and other low herbs" (translated from Dutch by M.S.). The sample of 1951 is labelled, "Road south of Voorst. Opposite "Empe"". We conclude that both samples were taken at the exact same locality.

(3) Allemansgeest. Shells collected by WC Van Heurn in summer 1942 ($N = 500$) and 1943 ($N = 772$) on the $150 \times 50$ m peninsula "Allemansgeest" at the confluence of Vliet and Korte Vliet, near Voorschoten (52.143°N 4.468°E). The location and samples were reported on by *Van Heurn (1943)* and *Van Heurn (1945)*. He describes the habitat as "a narrow spit of land, planted with willow, ash, and some alder, among which a wild herb vegetation appears in summer, consisting mostly of nettles" (translated from Dutch by M.S.). He also reports on a set of samples from 1941, which we could not locate. The collected numbers reported for 1942 and 1943 were, respectively, 535 and 787, suggesting that a small number of individuals had been lost from the samples. We assume these to have been random subsets.

(A fourth set, from the vicinity of Eenrum, was already reported on previously; *Ożgo & Schilthuizen, 2012*).

### Resampling in 2010

We visited these exact locations in 2010. All sites were identifiable and still contained *Cepaea nemoralis*. At Lobith, the vegetation remained as described by *Wolda (1969)*. Since snail densities were apparently much lower than in Wolda's time, we did not sample Wolda's individual subsections, but instead pooled all material from the entire 400-m length of dike (20–22 June 2010). At Empe, no patch of vegetation exactly matched the description by Van Heurn, so we sampled from two different patches facing the façade of the manor house: Empe-1 (52.1457°N 6.1435°E; reeds, nettles; 20–22 June 2010) and Empe-2 (52.1456°N 6.1426°E; maple forest; 20–22 June 2010). At Allemansgeest, finally, the vegetation had become more park-like, with lawns, tall poplars, and only a few patches where a herbaceous layer was present. *Cepaea* densities were low, so we sampled multiple times, on 21 June, 22 August, and 13 September 2010. At all sites, we collected juveniles and adults, (fresh) dead and alive. Juveniles too small to judge the colour morph accurately were not included. Fieldwork was conducted under permission FF/75A/2010/021a from the Netherlands Ministry of Agriculture, Nature, and Food Quality.

### Shell morph scoring scheme

Each individual from the museum samples and the recent samples was scored following the same scoring scheme as described in *Cain & Sheppard (1954)*. We investigated three types of changes in shell morph frequencies between the historical baseline data and the resampling data: (1) changes in shell ground colour: yellow (Y), pink (P), and brown (B); (2) changes in banding categories: mid-banded (00300; M), three-banded (00345; T), five-banded (12345; F), and other banding (O); and (3) changes in yellow effectively unbanded (YeU), the aggregate phenotype that includes all yellow shells with two upper bands missing (Y00XXX).

### Data analysis

At each site, and for each pair of samples, we performed separate chi-square homogeneity tests for each of the three types of morph change (banding, colour, and proportion of YeU) to examine whether there is evidence of a change in frequency between years (Fig. 1). Although we recorded the frequency of all morph categories, we excluded categories from the homogeneity tests when the morph frequency in the baseline year was equal to zero. The significance level of the chi-square test was set at $p = 0.05$, and we tested for the following changes in frequency between the baseline and resampling years. Colour: yellow, pink, and brown; shell banding: mid-banded, three-banded and five-banded; YeU: Yellow-effectively unbanded. All the analyses were calculated in Excel (Microsoft) (Data S1).

## RESULTS

Full results are given in Fig. 1 and Data S1–S4. We found no significant differences in colour ($\chi^2 = 2.38$, $d.f. = 2$, $p = 0.30$) or banding morph frequencies ($\chi^2 = 2.03$, $d.f. = 2$, $p = 0.36$) between the two patches sampled at Empe in 2010, so we pooled the data. Altogether, our 2010 sample sizes for Lobith, Empe, and Allemansgeest were $N = 131$, $N = 189$, and $N = 77$, respectively (Data S2–S4). Parts of these samples have
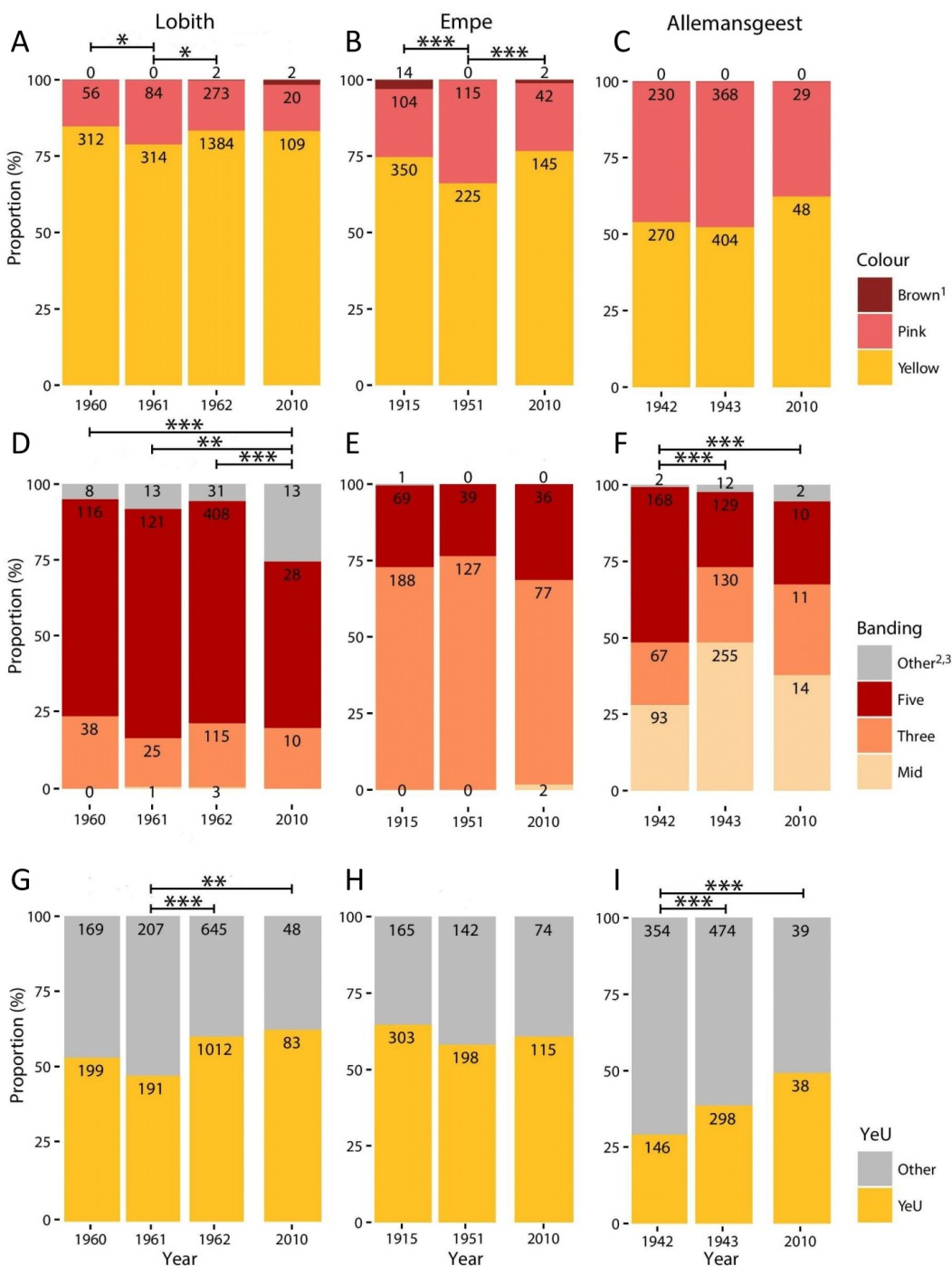

**Figure 1 Proportional variation in morph frequencies for all sites in all years.** (A–C), Shell ground colour morphs. (D–F), Banding categories —Mid, mid-banded (00300); Three, three-banded (00345); Five, five-banded (12345); Other, other banding categories. (G–I) Proportion of Yellow effectively unbanded (YeU). Numbers indicate total count. *, $p < 0.05$; **, $p < 0.01$; ***, $p < 0.001$.

been stored as vouchers in the alcohol collection of Naturalis Biodiversity Center, under collection numbers RMNH.5004222 (Allemansgeest), RMNH.5004223-5004224 (Lobith) and RMNH.5004225-5004227 (Empe).

In Lobith, significant changes in the frequencies of colour, banding, and the yellow effectively unbanded (YeU) phenotype occurred over the study period (Fig. 1). There was a shift in the frequency of yellow shells from 85% in 1960 to 79% in 1961 ($\chi^2 = 4.4$, $p < 0.05$), and then to 84% in 1962 ($\chi^2 = 4.8$, $p < 0.05$). Roughly the same frequency of yellow (83%) was recorded in 2010. The banding pattern was significantly different in 2010 as compared to the years 1960–1962 (1960–2010: $\chi^2 = 18.5$, $p < 0.001$; 1961–2010: $\chi^2 = 12.5$, $p < 0.01$; 1962–2010: $\chi^2 = 28.1$, $p < 0.001$), which included a decrease in the frequency of five-banded from 73% in 1962 to 55% in 2010, and an increase in the frequency of usually rare "other banding" snails, in this case the 00045 form, from 6% in 1962 to 26% in 2010. Between 1961 and 1962, the frequency of the YeU phenotype increased from 48% to 61% ($\chi^2 = 22.6$, $p < 0.001$). This was due to an increase in the frequency of yellow shells and a simultaneous increase in the frequency of yellow unbanded among yellow shells form 52% to 64%. Although there were some reversals, together these changes were in the direction of higher frequencies of less heavily banded forms.

In Empe, the population remained relatively stable between samples, with no significant differences in banding patterns or the frequency of YeU. Interestingly, data from 1951 suggest that some shifts in morph frequencies were occurring over that time despite the fact that the 1915 and 2010 samples do not differ significantly: the frequency of yellow in 1951 was lower (66%) ($\chi^2 = 7.1$, $p < 0.01$ and $\chi^2 = 6.4$, $p < 0.05$, respectively). This corresponded with an increase in the frequency of pink shells in 1951 from 22% to 34%.

In Allemansgeest, a significant change in the frequency of banding morphs occurred between 1942 and 1943 ($\chi^2 = 66.4$, $p < 0.001$), including an increase in the mid-banded from 28% to 48%, and a decrease in five-banded from 51% to 24%. The frequency of YeU also changed over this time period from 29% in 1942 to 39% in 1943 ($\chi^2 = 12.5$, $p < 0.001$). Colour morphs did not change significantly between those years; the frequency of yellow did increase from 52% to 62% between 1943 and 2010 ($\chi^2 = 2.8$, n.s.). As in Lobith, the overall change was in the direction of lighter shells.

## DISCUSSION

The use of natural history museum collections allowed us to analyse evolutionary changes in populations of *Cepaea nemoralis* that occurred over several decades at three sites in the Netherlands. Our study contributes to the increasing recognition that large changes in the states of populations can occur over very short time scales and exceed in magnitude the long-term trends (*Hendry & Kinnison, 1999*; *Siepielski, DiBattista & Carlson, 2009*). They may result from natural selection in response to biotic or abiotic factors (e.g., changing weather conditions and/or fluctuating selection by predators), from migration, and from random events. The data available in this study do not allow us to draw conclusions about the relative importance of those factors. Historical data from consecutive years show that the changes in the genetic structure of populations were rapid, but as the years

of sampling were different in the three sets of data, it remains unknown whether the recorded changes were consistent over different sites. In the long time-scale, the data from Lobith and Allemansgeest suggest that the general direction of changes was towards higher frequencies of lighter morphs (yellow and less heavily banded), which is consistent with the direction of selection predicted on the basis of the on-going climate change, and this has been shown in some other studies on *Cepaea* snails (*Ożgo & Schilthuizen, 2012*; *Cameron, Cook & Greenwood, 2013*; *Silvertown et al., 2011*). The value of the present study lies in the insight we were able to gain into changes in the genetic composition of populations over long time intervals. However, our interpretation is limited by the number of time points that the collections we studied could provide.

Thus, an additional reflection from our study concerns the value of long time-series of population samples (lots) maintained in museum collections. The importance of natural history museum collections is increasingly recognized (*Schilthuizen et al., 2015*; *Turney et al., 2015*; *Holmes et al., 2016*) but it is realistic to assume that for reasons of space, resources, and staff, their scope will remain limited, and careful choices of the focus of the on-going and future collection efforts need to be taken. We advocate especially for the collection of time-series of the (often-ignored) common species, with the view of securing future access to the records of population states of currently widespread species. Such collections have the potential of being analysed with the tools and resolution not yet available today, and being used to answer questions which at present are not even anticipated. Current examples of such applications include the vast field of museum genomics (*Yeates, Zwick & Mikheyev, 2016*), studies of amphibian chytridiomycosis (*James et al., 2015*), or of retroviral integration sites (*Cui et al., 2016*). Focusing on common species has the advantage of allowing to collect relatively large samples from designated populations at regular intervals. An important reason to focus on common species is that the present biodiversity crisis concerns many species that until recently were common and widespread, but are now in decline (*Gaston & Fuller, 2007*; *Inger et al., 2015*; *Petrovan & Schmidt, 2016*). Time-series collections can help document demographic, genetic, and evolutionary processes in populations undergoing distribution shifts, declines, and possibly rebounds. It is the common species with relatively high population numbers that are most likely to undergo evolutionary rescue (*Carlson, Cunningham & Westley, 2014*), and time-series collections have the potential of capturing its occurrence in wild populations.

## ACKNOWLEDGEMENTS

We thank Jeroen Goud and Bram van der Bijl, collection managers of the Naturalis Mollusca collection for providing access to the historical specimens and for arranging the administration of the 2010 vouchers. Henrik Wolda, Rinny Kooi, Wilke van Delden, and Lucy Oosterhoff helped in obtaining additional information on historical collection localities. Laurence Cook, Kerstin Johannesson, and Andrew Hendry reviewed an earlier version of this paper, and their comments helped us to rewrite and improve it.

### Funding
The authors received no funding for this work.

### Competing Interests
The authors declare there are no competing interests.

### Author Contributions

- Małgorzata Ożgo conceived and designed the experiments, performed the experiments, analyzed the data, contributed reagents/materials/analysis tools, reviewed drafts of the paper.
- Thor-Seng Liew performed the experiments, analyzed the data, contributed reagents/materials/analysis tools, reviewed drafts of the paper.
- Nicole B. Webster performed the experiments, analyzed the data, contributed reagents/materials/analysis tools, prepared figures and/or tables, reviewed drafts of the paper.
- Menno Schilthuizen conceived and designed the experiments, performed the experiments, wrote the paper, prepared figures and/or tables.

### Field Study Permissions
The following information was supplied relating to field study approvals (i.e., approving body and any reference numbers):

Field work was carried out under permission FF/75A/2010/021a from the Netherlands Ministry of Agriculture, Nature, and Food Quality.

### Data Availability
The raw data has been supplied as a Supplementary File.

### Supplemental Information
Supplemental information for this article can be found online at http://dx.doi.org/10.7717/peerj.3938#supplemental-information.

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
