# Peer review of "Inferring microevolution from museum collections and resampling: lessons learned from Cepaea"

_PeerJ, doi:10.7717/peerj.3938_

## Round 0.1 · original submission · Major Revisions

In this manuscript the authors compare the frequencies of genetically based shell characteristics in several temporally sampled populations of the land snail Cepaea nemoralis. The samples of populations span decades to almost a century, and the primary conclusion is that year-to-year fluctuations are of great enough magnitude that it could be problematic if one were to infer the magnitude or direction of long-term evolutionary trends from the comparison of two time points. The reviewers all found the basic experimental design valid and appropriate. They differed somewhat on the interpretation of results and the best focus of the discussion. It is one of the most fundamental tenets of evolutionary theory that organisms respond to their current environment, not unknowable future environments so if conditions fluctuate on short or medium term time scales, populations responses to these conditions will fluctuate as conditions fluctuate. So, I am in agreement with reviewers 2 and 3, that this is a pretty obvious conclusion supported by both long term studies such as the Grants work on Darwin’s finches, and lots of paleontological studies. I agree with reviewer 3 that it makes more sense to cast this as a noise versus signal issue. The magnitude of short term changes may make it very difficult to infer long-term trends as the noise of short term fluctuations may mask long-term trends (See figure 2 of Hendry and Kinnison, 1999, as noted by reviewer 3). I encourage the authors to recast this as suggested by reviewer 3, or at the very least give a robust sampling of publications where authors make the error they are suggesting by inferring long-term trends from a two-point sampling in studies of inferred recent rapid evolutionary change. Finally, as noted by reviewer one discuss possible confounding issues, immigration or emigration to sites, variable reproductive success, collecting bias, etc.

Note that Cook, Cowie, and Jones, 1999 is missing from references.

·

Basic reporting

This is a clearly presented and interesting account of changes in recorded morph frequencies in the snail Cepaea nemoralis over time. The animal is an important model for the study of changes, rapid or longer term, in response to changing environment. The evidence for rapid response and short-term fluctuation is an area of current interest and this paper is a worthy addition to it.

I do, however, think that the authors need to think more fully about the conclusions from the data and make a much more forceful case for their interpretation if they wish to stand by it.

Experimental design

Experimental design is clearly stated and appropriate to the objectives of the analysis.

Validity of the findings

The data are robust and statistically sound and the analysis appropriate to meet the objectives set out. I have some issues with the interpretation, which will be set out in the next box.

Additional comments

Comments from Laurence Cook.

From my own experience of the subject of your paper I have the following comments to make on the conclusions.

Line 55. ff. The authors will know that I have reservations about the treatment of eU as a genetic category. Although Cain and Sheppard (line 206. particularly Cain, who emphasised the fact that this was a visual phenotype) noted the predictive value of eU in relation to habitat, analysis of further data suggests that it is no improvement of the category unbanded itself (Cameron & Cook. 2012. Folia Malacologia 20, 255-263). It would be safer to refer to the change as adaptive rather than genetic.

Line 61. ff. Another important long-term study, which appears to show stability and also considers movement and one of the selective agents, is by Charles Goodhart ( 1956 Proc. Linnean Society of London 167, 50-67; 1956 Proc. Linnean Society of London 169, 163-167.

Line 127. ff. The sample sizes for Lobith are colossal, surely much larger than a possible effective population? I have not gone back to Wolda’s paper, but is there a possibility that some frequencies quoted are ensemble values whereas with the later smaller densities you would have been examining sections? I think a note on this possibility should be added.

Line 153. Does that imply that some morph or morphs entered the samples after the first sampling?

Line 218. ff. Surely the caution should be applied to the short-term states not the long-term states (see below)?

Line 250. ff. But patterns can be perceived, and are very much dependent on one’s confidence that the appropriate populations are being sampled at the different times (Cameron & Cook. 2012. Folia Malacologia 20, 255-263; Cook. 2013. J. Moll. Stud. 80, 43-46).

General. I think it is important to investigate and interpret the results you describe. Nevertheless, the problem I have with the interpretation is in your sentence beginning line 202. If short term fluctuations characteristically exceed longer term change, apparently sometimes by large amounts, the implication must be that some kind of stabilizing force also operates. Otherwise fluctuations would accumulate and lead to monomorphism or extinction. Why does this not happen? We carried out a very detailed analysis of the data looked at by Bell (Cain et al., 1990) and in the end concluded that there was no evidence of forces shifting frequencies on a yearly basis. The estimations of ingress and egress did not show up age classes suffering from high mortality or radical difference in contribution of morph types to a particular recruitment. The alternative would have to be that, for whatever reason, the methods or circumstances of sampling sometimes introduce biases. At Point of Air (Cook & Pettitt, 1998. Biol. J. Linn. Soc. 64, 137-150) there is no evidence of selection introduced by climate etc. but plenty of reason to suppose that extensive movement of individuals took place that could account for what would look like a selective change if sampling was restricted to a limited part. In another sampling (Cook 2003. J. Conchology 38, 73-78) I also noted a change in frequency in the middle of a three-sample (in fact, a four-sample as one of yours) sequence which later appeared to be cancelled. A possible reason discussed could have been that when density was low relative visibility of morphs changed. I think that at least you need to discuss these problems more fully before concluding that frequencies do in fact change radically on a very short-term (sometimes apparently within-generation) basis.

·

Basic reporting

This short manuscript is right to the point with an accurate and relevant message. However, I found the message so trivial that it should not be necessary to point out these things. So, what I should like of the authors is that they give more meat on the bones when referring to the older literature and give very detailed examples of when authors has drawn conclusions of long-term trends from just two data points (one historical and one contemporary).
Moreover, it would be interesting to learn if this is the case for colour-variation in Cepaea snails, usually, or if similar inferences have been drawn for other traits that are tentatively under selection. Examples would be welcome.
Finally, I would suggest yet another possible way of assessing interannual variation in order to separate this from long-term trends if only one historical sample is available: Instead of using only one year of contemporary sampling, repeat sampling over 2-3 contemporary years to assess interannual variation.

Experimental design

The experimental design is straight forward and only basic statistics is used.

Validity of the findings

I wish I could conclude that this information is of very little value because conclusions of long-term trends are never drawn from only two data points. But, if this is so (and as suggested above, I think the authors must do more to show that this is actually the case) then this study has some value. In particular, if the authors (as is partly done) suggest approaches to solve the problem.
However, if the authors fail to find good example of studies in which conclusions are based on only one historical and one contemporary sample, then I think this study has only very minor value, as this should be rather obvious to anyone.

Additional comments

My main point is that the authors should put more emphasize on examples of studies in which this has been a problem. It would also be good to focus the study more on how the problem can be solved. There is some discussion about this but this part can be more of a focus in both Abstract, Introduction and aim of the study. How do you deal with a situation when there is one historical data point available....
Such a focus would be much more of a contribution to the scientific society than only pointing to an obvious problem.

·

Basic reporting

Please see general comments.

Experimental design

Please see general comments.

Validity of the findings

Please see general comments.

Additional comments

The authors examine how the frequencies of genetically-based color morphs of Cepaea change from historical samples to the present, paying special attention to how conclusions can change depending on which specific year from a given time period is used for a comparison between time periods. In essence, conclusions as to how frequencies change from the past the present depend on which specific year from the past is used for comparison. The implication is that studies need to have replicate samples from each time period if they are to generate reliable estimates of change BETWEEN time periods.

This is all true but not especially surprising, as long term data sets (e.g., Darwin’s finches) show tons of year-to-year variation despite the lack of long term trends. I suppose there isn’t any harm in restating the point for another study system but the insight isn’t particularly novel (see also fig 2 of Hendry and Kinnison 1999). I do object to how the implication is sometimes stated as the lack of replicate sampling leading to incorrect or “misleading” inferences. Instead (as is implied at other parts of the MS), the robust conclusion is that short term evolutionary changes can be greater than long term changes. I would prefer that to be the point of motivation for the paper rather than the more obvious and less interesting point that somehow “incorrect” conclusions can result from incomplete sampling. Instead, the observed changes are “correct” but they don’t necessarily reveal long term average trends. Thus, the problem isn’t in how the comparison is done but rather in how an investigator might attempt to extend a particular temporal comparison to general statements about long-term trends. This is perhaps a subtle point but the wording in the MS could be much improved in this regard.

Technical comments:
1. Line 38: cite general meta-analyses and quantitative reviews rather than a few arbitrary examples out of hundreds that could be cited.
2. Line 45: Of course, this point is obvious from essentially any long-term study.
3. The emphasis seems to be on the fact that SIGNIFICANCE of temporal changes depends on year of sampling. However, significance is not what matters here (given that significance is also dependent on sample size). Instead, it is effect size that matters – that is, the actual frequency change. So the real question is how much allele frequency change differs depending on the year of sampling. Of course, these data are in the figure but the point is not clear in the text, nor is the figure optimal. For best inference, maybe prepare a graph of “change using one historical year” on the x-axis and “change using another historical year” on the y-axis. Deviations from the 1:1 line then reveal the extent to which the chosen year matters.
4. Also, the allele frequency change that can occur is dependent on the starting allele frequency, so it would be good to examine if the two are correlated.
5. As the lead author has shown in other work, the multivariate phenotypes of snails can vary among similar habitats. I would like to see a graph something like Fig 3 of Ozgo et al. (2011 – BJLS 102:251-262).
6. The starting part of the Discussion essentially just lists the results again. I think the discussion could be much more interesting by focusing on the processes that shape short-term versus long-term change and how these process influence expectations under climate change (or something like that).

---

## Round 0.2 · accepted · Accept

In this revised manuscript, the authors compare the frequencies of genetically-based shell characteristics in several temporally sampled populations of the land snail Cepaea nemoralis, a model system for studying local adaption. The samples of populations span decades to almost a century, and the primary conclusion is short-term changes in populations are of greater magnitude than some long-term trends, complicating the analysis of rapid evolutionary changes that may be associated with human-induced environmental effects. Three reviewers of an earlier draft of the manuscript thought that the data were valuable and properly analyzed, but thought that the discussion had been cast in such a way that the answer seemed obvious. The intro and discussion have now been recast to no longer focus primarily on the potentially misleading effects of two-point comparisons, and instead emphasizes the magnitude of short-term changes relative to long-term trends, and the need for multiple temporal points to draw valid conclusions about long-term trends. Other reviewer's points have also been addressed, and, in addition, the study makes good use of historical museum collections, making it of more general interest. I consider the manuscript now ready for publication.